# LRM: LARGE RECONSTRUCTION MODEL FOR SINGLE IMAGE TO 3D

**Yicong Hong**[1,2]* **Kai Zhang**[1] **Jiuxiang Gu**[1] **Sai Bi**[1] **Yang Zhou**[1]
**Difan Liu**[1] **Feng Liu**[1] **Kalyan Sunkavalli**[1] **Trung Bui**[1] **Hao Tan**[1]

[1]Adobe Research    [2]Australian National Univeristy

mr.yiconghong@gmail.com
{kaiz,jigu,sbi,yazhou,diliu,fengl,sunkaval,bui,hatan}@adobe.com

## ABSTRACT

We propose the first Large Reconstruction Model (LRM) that predicts the 3D model of an object from a single input image within just 5 seconds. In contrast to many previous methods that are trained on small-scale datasets such as ShapeNet in a category-specific fashion, LRM adopts a highly scalable transformer-based architecture with 500 million learnable parameters to directly predict a neural radiance field (NeRF) from the input image. We train our model in an end-to-end manner on massive multi-view data containing around 1 million objects, including both synthetic renderings from Objaverse and real captures from MVImgNet. This combination of a high-capacity model and large-scale training data empowers our model to be highly generalizable and produce high-quality 3D reconstructions from various testing inputs, including real-world in-the-wild captures and images created by generative models. Video demos and interactable 3D meshes can be found on our LRM project webpage: https://yiconghong.me/LRM.

## 1 INTRODUCTION

Imagine if we could instantly create a 3D shape from a single image of an arbitrary object. Broad applications in industrial design, animation, gaming, and AR/VR have strongly motivated relevant research in seeking a generic and efficient approach towards this long-standing goal. Due to the underlying ambiguity of 3D geometry in a single view, early learning-based methods usually perform well on specific categories, utilizing the category data prior to infer the overall shape (Yu et al., 2021). Recently, advances in image generation, such as DALL-E (Ramesh et al., 2021) and Stable Diffusion (Rombach et al., 2022), have inspired research that leverages the remarkable generalization capability of 2D diffusion models to enable multi-view supervision (Liu et al., 2023b; Tang et al., 2023). However, many of these methods require delicate parameter tuning and regularization, and their results are limited by the pre-trained 2D generative models. Meanwhile, there are many approaches that rely on per-shape optimization (*e.g.* optimize a NeRF (Mildenhall et al., 2021; Chan et al., 2022; Chen et al., 2022a; Müller et al., 2022; Sun et al., 2022)) to construct a consistent geometry; this process is often slow and impractical.

On the other hand, the great success in natural language processing (Devlin et al., 2018; Brown et al., 2020; Chowdhery et al., 2022) and image processing (Caron et al., 2021; Radford et al., 2021; Alayrac et al., 2022; Ramesh et al., 2022) can be largely credited to three critical factors: (1) using highly scalable and effective neural networks, such as the Transformers (Vaswani et al., 2017), for modeling the data distribution, (2) enormous datasets for learning generic priors, as well as (3) self-supervised-like training objectives that encourage the model to discover the underlying data structure while maintaining high scalability. For instance, the GPT (generative pre-trained transformer) series (Radford et al., 2019; Brown et al., 2020; OpenAI, 2023) build large language models with huge transformer networks, large-scale data, and the simple next-word prediction task. In light of this, we pose the same question for 3D: given sufficient 3D data and a large-scale training framework, ***is it possible to learn a generic 3D prior for reconstructing an object from a single image?***

---

*Intern at Adobe Research.

In this paper, we propose a **L**arge **R**econstruction **M**odel (LRM) for single-image to 3D. Our method adopts a large transformer-based encoder-decoder architecture for learning 3D representations of objects from a single image in a data-driven manner. Our method takes an image as input and regresses a NeRF in the form of a triplane representation (Chan et al., 2022). Specifically, LRM utilizes the pre-trained visual transformer DINO (Caron et al., 2021) as the image encoder to generate the image features, and learns an image-to-triplane transformer decoder to project the 2D image features onto the 3D triplane via cross-attention and model the relations among the spatially-structured triplane tokens via self-attention. The output tokens from the decoder are reshaped and upsampled to the final triplane feature maps. Afterwards, we can render the images at an arbitrary view by decoding the triplane feature of each point with an additional shared multi-layer perception (MLP) to get its color and density and performing volume rendering.

The overall design of LRM maintains high scalability and efficiency. In addition to the use of a fully transformer-based pipeline, a triplane NeRF is a concise and scalable 3D representation since it is computationally friendly compared to other representations such as volumes and point clouds. It also has a better locality with respect to the image input compared to tokenizing the NeRF's model weights as in Shap-E (Jun & Nichol, 2023). Moreover, our LRM is trained by simply minimizing the difference between the rendered images and ground truth images at novel views, without excessive 3D-aware regularization or delicate hyper-parameter tuning, allowing the model to be very efficient in training and adaptable to a wide range of multi-view image datasets.

To the best of our knowledge, LRM is the first *large-scale 3D reconstruction model*; it contains more than 500 million learnable parameters, and it is trained on approximately one million 3D shapes and video data across diverse categories (Deitke et al., 2023; Yu et al., 2023); this is substantially larger than recent methods that apply relatively shallower networks and smaller datasets (Chang et al., 2015; Reizenstein et al., 2021; Downs et al., 2022). Through experiments, we show that LRM can reconstruct high-fidelity 3D shapes from a wide range of images captured in the real world, as well as images created by generative models. LRM is also a highly practical solution for downstream applications since it can produce a 3D shape in just five seconds[1] without post-optimization.

## 2 Related work

**Single Image to 3D Reconstruction**   Extensive efforts have been devoted to address this problem, including early learning-based methods that explore point clouds (Fan et al., 2017; Wu et al., 2020), voxels (Choy et al., 2016; Tulsiani et al., 2017; Chen & Zhang, 2019), and meshes (Wang et al., 2018; Gkioxari et al., 2019), as well as various approaches that learn implicit representations such as SDFs (Park et al., 2019; Mittal et al., 2022), occupancy networks (Mescheder et al., 2019), and NeRF (Jang & Agapito, 2021; Müller et al., 2022). Leveraging 3D templates (Roth et al., 2016; Goel et al., 2020; Kanazawa et al., 2018; Kulkarni et al., 2020), semantics (Li et al., 2020), and poses (Bogo et al., 2016; Novotny et al., 2019) as shape priors have also been widely studied in category-specific reconstruction. Category-agnostic methods show great generalization potential (Yan et al., 2016; Niemeyer et al., 2020), but they often unable to produce fine-grained details even when exploiting spatially-aligned local image features (Xu et al., 2019; Yu et al., 2021).

Very recently, there is an emerging trend of using pre-trained image/language models (Radford et al., 2021; Li et al., 2022; 2023b; Saharia et al., 2022; Rombach et al., 2022), to introduce semantics and multi-view guidance for image-to-3D reconstruction (Liu et al., 2023b; Tang et al., 2023; Deng et al., 2023; Shen et al., 2023b; Anciukevičius et al., 2023; Melas-Kyriazi et al., 2023; Metzer et al., 2023; Xu et al., 2023; Qian et al., 2023; Li et al., 2023a). For instance, Zero-1-to-3 fine-tunes the Stable Diffusion model to generate novel views by conditioning on the input image and camera poses (Liu et al., 2023b); its view consistency and reconstruction efficiency have been further improved by Liu et al. (2023a). Make-It-3D (Tang et al., 2023) uses BLIP to generate text descriptions for the input image (which is applied to guide the text-to-image diffusion) and trains the model with score distillation sampling loss (Poole et al., 2022) and CLIP image loss to create geometrically and semantically plausible shapes.

---

[1]Five seconds per shape on a single NVIDIA A100 GPU, including around 1.14 seconds image-to-triplane feed-forward time, 1.14 seconds to query resolution of $384 \times 384 \times 384$ points from the triplane-NeRF, and 1.91 seconds mesh extraction time using Marching Cubes (Lorensen & Cline, 1998).

In contrast to all these methods, our LRM is a purely data-driven approach that learns to reconstruct arbitrary objects in the wild. It is trained with minimal and extensible 3D supervision (*i.e.*, rendered or captured 2D images of 3D objects) and does not rely on any guidance from pre-trained vision-language contrastive or generative models.

**Learning 3D Representations from Images** 3D reconstruction from a single image is an ill-posed problem that has been frequently addressed by models with generative properties. Many previous works apply an encoder-decoder framework to model the image-to-3D data distribution (Choy et al., 2016; Yan et al., 2016; Dai et al., 2017; Xu et al., 2019; Wu et al., 2020; Müller et al., 2022; Sajjadi et al., 2022; Goel et al., 2023), where a compact latent code is trained to carry the texture, geometry, and pose details of the target. However, learning such an expressive representation usually requires a capable network and abundant 3D data which is very expensive to acquire. Hence most of these methods only focus on a few categories and produce very coarse results. GINA-3D (Shen et al., 2023a) implements a model that applies a visual transformer encoder and cross-attention (instead of a transformer decoder as in LRM) to translate images to triplane representations. However, the model and training are much smaller in scale, and their work has a different focus on category-specific generation. Recent data-driven approach MCC (Wu et al., 2023) trains a generalizable transformer-based decoder with CO3D-v2 data (Reizenstein et al., 2021) to predict occupancy and color from the input image and its unprojected point cloud. Although MCC can handle real and generated images and scenes, the results are usually over-smooth and lose details.

**Multimodal 3D** Motivated by the great advances in 2D multimodal learning (Tan & Bansal, 2019; Chen et al., 2020; 2022b; Yu et al., 2022; Singh et al., 2022; Wang et al., 2022; Alayrac et al., 2022; Girdhar et al., 2023), LRM considers 3D as a new modality and directly grounds 2D feature maps onto 3D triplane via cross-attention. There are early attempts in this direction that minimize the difference between encoded image and 3D representations (Girdhar et al., 2016; Mandikal et al., 2018), as well as recent research, ULIP (Xue et al., 2023) and CLIP[2] (Zeng et al., 2023), which bridges 3D, language, and images via contrastive learning. LERF (Kerr et al., 2023) learns a language field inside NeRF by rendering CLIP embeddings along training rays. In contrast, our method focuses on generic single image-to-3D reconstruction. We would like to mention the concurrent work Cap3D (Luo et al., 2023) that produces descriptions for 3D shapes by applying BLIP (Li et al., 2023b) to generate captions of different views, uses GPT-4 (OpenAI, 2023) to summarize them, and then employs these language-3D pairs for training text-to-3D generative models (Nichol et al., 2022; Poole et al., 2022; Jun & Nichol, 2023). There are also recent works in connecting 3D and large language models, such as 3D-LLM (Hong et al., 2023) and LLM-Grounder (Yang et al., 2023).

## 3 METHOD

In this section, we detail the proposed LRM architecture (Fig. 1). LRM contains an image encoder that encodes the input image to patch-wise feature tokens (Sec. 3.1), followed by an image-to-triplane decoder that projects image features onto triplane tokens via cross-attention (Sec. 3.2). The output triplane tokens are upsampled and reshaped into the final triplane representation, which is used to query 3D point features. Lastly, the 3D point features are passed to a multi-layer perception to predict RGB and density for volumetric rendering (Sec. 3.3). The training objectives and data are described in Sec. 3.4 and Sec. 4.1.

### 3.1 IMAGE ENCODER

Given an RGB image as input, LRM first applies a pre-trained visual transformer (ViT) (Dosovitskiy et al., 2020) to encode the image to patch-wise feature tokens $\{\boldsymbol{h}_i\}_{i=1}^n \in \mathbb{R}^{d_E}$, where $i$ denotes the $i$-th image patch, $n$ is the total number of patches, and $d_E$ is the latent dimension of the encoder. Specifically, we use DINO (Caron et al., 2021), a model trained with self-distillation that learns interpretable attention over the structure and texture of the salient content in images. Compared to other semantic-oriented representations such as the visual features from ImageNet-pretrained ResNet (He et al., 2016) or CLIP (Radford et al., 2021), the detailed structural and texture information in DINO is more important in our case since LRM can use it to reconstruct the geometry and color in 3D space.

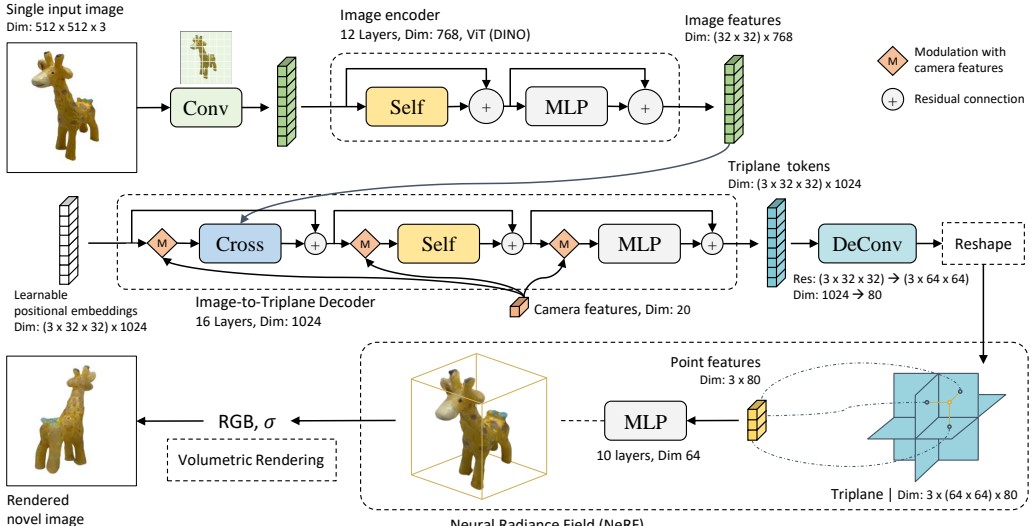

Figure 1: The overall architecture of LRM, a fully-differentiable transformer-based encoder-decoder framework for single-image to NeRF reconstruction. LRM applies a pre-trained vision model (DINO) to encode the input image (Sec. 3.1), where the image features are projected to a 3D triplane representation by a large transformer decoder via cross-attention (Sec. 3.2), followed by a multi-layer perceptron to predict the point color and density for volumetric rendering (Sec. 3.3). The entire network is trained end-to-end on around a million of 3D data (Sec. 4.1) with simple image reconstruction losses (Sec. 3.4).

As a result, instead of only using the ViT pre-defined class token `[CLS]` that aggregates patch-wise features, we also utilize the entire feature sequence $\{\boldsymbol{h}_i\}_{i=1}^n$ to better preserve this information[2].

## 3.2 IMAGE-TO-TRIPLANE DECODER

We implement a transformer decoder to project image and camera features onto learnable spatial-positional embeddings and translate them to triplane representations. This decoder can be considered as a prior network that is trained with large-scale data to provide necessary geometric and appearance information to compensate for the ambiguities of single-image reconstruction.

**Camera Features** We construct the camera feature $\boldsymbol{c} \in \mathbb{R}^{20}$ of the input image by flattening out the 4-by-4 camera extrinsic matrix $\boldsymbol{E}$ (that represents the camera-to-world transformation) and concatenate it with the camera focal length $foc$ and principal point $pp$ as $\boldsymbol{c} = [\boldsymbol{E}_{1 \times 16}, foc_x, foc_y, pp_x, pp_y]$. Moreover, we normalize the camera extrinsic $\boldsymbol{E}$ by similarity transformations so that all the input cameras are aligned on the same axis (with the lookup direction aligned with the $z$-axis). Note that, LRM does not depend on a canonical pose of the object, and the ground truth $\boldsymbol{c}$ is only applied in training. Conditioning on normalized camera parameters greatly reduces the optimization space of triplane features and facilitates model convergence (see details in Sec. 4.2). To embed the camera feature, we further implement a multi-layer perceptron (MLP) to map the camera feature to a high-dimensional camera embedding $\tilde{\boldsymbol{c}}$. The intrinsics (focal and principal point) are normalized by the image's height and width before sending to the MLP layer.

**Triplane Representation** We follow previous works (Chan et al., 2022; Gao et al., 2022) to apply triplane as a compact and expressive feature representation of the reconstruction subject. A triplane $\boldsymbol{T}$ contains three axis-aligned feature planes $\boldsymbol{T}_{XY}$, $\boldsymbol{T}_{YZ}$ and $\boldsymbol{T}_{XZ}$. In our implementation, each plane is of dimension $(64 \times 64) \times d_T$ where $64 \times 64$ is the spatial resolution, and $d_T$ is the number of feature channels. For any 3D point in the NeRF object bounding box $[-1, 1]^3$, we can project it onto each

---

[2]For simplicity, we use $\{\boldsymbol{h}_i\}_{i=1}^n$ in the following to denote the concatenated sequence of the encoded `[CLS]` token and patch-wise features.

of the planes and query the corresponding point features $(\boldsymbol{T}_{xy}, \boldsymbol{T}_{yz}, \boldsymbol{T}_{xz})$ via bilinear interpolation, which is then decoded by an $\mathrm{MLP}^{nerf}$ into the NeRF color and density (Sec. 3.3).

To obtain the triplane representation $\boldsymbol{T}$, we define learnable spatial-positional embeddings $\boldsymbol{f}^{init}$ of dimension $(3{\times}32{\times}32){\times}d_D$ which guide the image-to-3D projection and are used to query the image features via cross-attention, where $d_D$ is the hidden dimension of the transformer decoder. The number of tokens in $\boldsymbol{f}^{init}$ is smaller than the number of final triplane tokens ($3{\times}64{\times}64$); we will upsample the output of the transformer $\boldsymbol{f}^{out}$ to the final $\boldsymbol{T}$. In the forward pass, conditioning on the camera features $\tilde{\boldsymbol{c}}$ and image features $\{\boldsymbol{h}_i\}_{i=1}^n$, each layer of our image-to-triplane transformer decoder gradually updates the initial positional embedding $\boldsymbol{f}^{init}$ to the final triplane features via modulation and cross-attention, respectively. The reason for applying two different conditional operations is that the camera controls the orientation and distortion of the whole shape, whereas the image features carry the fine-grained geometric and color information that need to be embedded onto the triplane. Details of the two operations are explained below.

**Modulation with Camera Features**  Our camera modulation is inspired by DiT (Peebles & Xie, 2022) which implements an adaptive layer norm (adaLN) to modulate image latents with denoising timesteps and class labels. Suppose $\{\boldsymbol{f}_j\}$ is a sequence of vectors in transformer, we define our modulation function $\mathrm{ModLN}_c(\boldsymbol{f}_j)$ with camera feature $\boldsymbol{c}$ as

$$\gamma, \beta = \mathrm{MLP}^{\mathrm{mod}}(\tilde{\boldsymbol{c}}) \tag{1}$$

$$\mathrm{ModLN}_c(\boldsymbol{f}_j) = \mathrm{LN}(\boldsymbol{f}_j) \cdot (1 + \gamma) + \beta \tag{2}$$

where $\gamma$ and $\beta$ are the scale and shift (Huang & Belongie, 2017) output by $\mathrm{MLP}^{\mathrm{mod}}$ and LN is the Layer Normalization (Ba et al., 2016). Such modulation is applied to each attention sub-layer which will be specified next.

**Transformer Layers**  Each transformer layer contains a cross-attention sub-layer, a self-attention sub-layer, and a multi-layer perceptron sub-layer (MLP), where the input tokens to each sub-layer are modulated by the camera features. Suppose feature sequence $\boldsymbol{f}^{in}$ is the input of an transformer layer, we can consider $\boldsymbol{f}^{in}$ as the triplane hidden features since they are corresponding to the final triplane features $\boldsymbol{T}$. As shown in the decoder part of Fig. 1, the cross-attention module firstly attends from the triplane hidden features $\boldsymbol{f}^{in}$ to the image features $\{\boldsymbol{h}_i\}_{i=1}^n$, which can help linking image information to the triplane. Note that we here do not explicitly define any spatial alignment between the 2D images and 3D triplane hidden features, but consider 3D as an independent modality and ask the model to learn the 2D-to-3D correspondence by itself. The updated triplane hidden features will be passed to a self-attention sub-layer that further models the intra-modal relationships across the spatially-structured triplane entries. Then, a multi-layer perceptron sub-layer ($\mathrm{MLP}^{tfm}$) follows as in the original Transformer (Vaswani et al., 2017) design. Lastly, the output triplane features $\boldsymbol{f}^{out}$ will become the input to the next transformer layer.

Such a design is similar to the Perceiver network (Jaegle et al., 2021) while our model maintains a high-dimensional representation across the attention layers instead of projecting the input to a latent bottleneck. Overall, we can express this process for each $j$-th triplane entry in each layer as

$$\boldsymbol{f}_j^{cross} = \mathrm{CrossAttn}(\mathrm{ModLN}_c(\boldsymbol{f}_j^{in}); \{\boldsymbol{h}_i\}_{i=1}^n) + \boldsymbol{f}_j^{in} \tag{3}$$

$$\boldsymbol{f}_j^{self} = \mathrm{SelfAttn}(\mathrm{ModLN}_c(\boldsymbol{f}_j^{cross}); \{\mathrm{ModLN}_c(\boldsymbol{f}_j^{cross})\}_j) + \boldsymbol{f}_j^{cross} \tag{4}$$

$$\boldsymbol{f}_j^{out} = \mathrm{MLP}^{tfm}(\mathrm{ModLN}_c(\boldsymbol{f}_j^{self})) + \boldsymbol{f}_j^{self} \tag{5}$$

The $\mathrm{ModLN}$ operators in sub-layers (*i.e.*, $\mathrm{CrossAttn}$, $\mathrm{SelfAttn}$, $\mathrm{MLP}^{tfm}$) use different set of learnable parameters in the layer normalization and the modulation $\mathrm{MLP}^{mod}$. We do not add additional superscript to differentiate them for clarity.

The transformer layers are processed sequentially. After all the transformer layers, we obtain the output triplane features $\boldsymbol{f}^{\mathrm{out}}$ from the last layer as the output of the decoder. This final output is upsampled by a learnable de-convolution layer and reshaped to the final triplane representation $\boldsymbol{T}$.

## 3.3   Triplane-NeRF

We employ the triplane-NeRF formulation (Chan et al., 2022) and implement an $\mathrm{MLP}^{nerf}$ to predict RGB and density $\sigma$ from the point features queried from the triplane representation $\boldsymbol{T}$. The

$\text{MLP}^{nerf}$ contains multiple linear layers with ReLU (Nair & Hinton, 2010) activation. The output dimension of the $\text{MLP}^{nerf}$ is 4 where the first three dimensions are RGB colors and the last dimension corresponds to the density of the field. We refer to the Appendix for the details of NeRF volumetric rendering.

## 3.4 TRAINING OBJECTIVES

LRM produces the 3D shape from a single input image and leverages additional side views to guide the reconstruction during training. For each shape in the training data, we consider $(V-1)$ randomly chosen side views for supervision; we apply simple image reconstruction objectives between the $V$ rendered views $\hat{\boldsymbol{x}}$ and the ground-truth views $\boldsymbol{x}^{GT}$ (include the input view and side views). More precisely, for every input image $\boldsymbol{x}$, we minimize:

$$\mathcal{L}_{\text{recon}}(\boldsymbol{x}) = \frac{1}{V}\sum_{v=1}^{V}\left(\mathcal{L}_{\text{MSE}}(\hat{\boldsymbol{x}}_v, \boldsymbol{x}_v^{GT}) + \lambda\mathcal{L}_{\text{LPIPS}}(\hat{\boldsymbol{x}}_v, \boldsymbol{x}_v^{GT})\right) \tag{6}$$

where $\mathcal{L}_{\text{MSE}}$ is the normalized pixel-wise L2 loss, $\mathcal{L}_{\text{LPIPS}}$ is the perceptual image patch similarity (Zhang et al., 2018) and $\lambda$ is a customized weight coefficient.

## 4 EXPERIMENTS

### 4.1 DATA

LRM relies on abundant 3D data from Objaverse (Deitke et al., 2023) and MVImgNet (Yu et al., 2023), consisting of synthetic 3D assets and videos of objects in the real world, respectively, to learn a generalizable cross-shape 3D prior. For each 3D asset in Objaverse, we normalize the shape to the box $[-1, 1]^3$ in world space and render 32 random views with the same camera pointing toward the shape at arbitrary poses. The rendered images are of resolution $1024 \times 1024$, and the camera poses are sampled from a ball of radius $[1.5, 3.0]$ and with height in range $[-0.75, 1.60]^3$. For each video, we utilize the extracted frames from the dataset. Since the target shape in those frames can be at random positions, we crop and resize all of them using the predicted object mask[4] so that the object is at the center of the resulting frames; we adjust the camera parameters accordingly. Note that our method does not model background, hence we render images from Objaverse with a pure white background, and use an off-the-shelf package[4] to remove the background of video frames. In total, we pre-processed 730,648 3D assets and 220,219 videos for training.

To evaluate the performance of LRM on arbitrary images, we collected novel images from Objaverse (Deitke et al., 2023), MvImgNet (Yu et al., 2023), ImageNet (Deng et al., 2009), Google Scanned Objects (Downs et al., 2022), Amazon Berkeley Objects (Collins et al., 2022), captured new images in the real world, and generated images with Adobe Firefly[5] for reconstruction. We visualize their results in Sec. 4.3.1 and Appendix. To numerically study the design choices of our approach, we randomly acquired 50 unseen 3D shapes from the Objaverse and 50 unseen videos from the MvImgNet dataset, respectively. For each shape, we pre-process 15 reference views and pass five of them to our model one by one to reconstruct the same object, and evaluate the rendered images using all 15 reference views (see analyses in Appendix).

### 4.2 IMPLEMENTATION DETAILS

**Camera Normalization** We normalize the camera poses corresponding to the input images to facilitate the image-to-triplane modeling. Specifically, for the images rendered from synthetic 3D assets in Objaverse, regardless of the corresponding positions of the cameras, we normalize the input camera poses to position $[0, -2, 0]$ with the camera vertical axis aligned with the upward $z$-axis in the world frame. For the video data, since the camera can be at an arbitrary distance from the target and the object is not at the image center, we only normalize the camera pose to $[0, -dis, 0]$ where $dis$ is the original distance between world origin and camera origin.

---

[3]Most of Objaverse assets have consistent $z$-axis up.

[4]Rembg package, a tool to remove image background: https://pypi.org/project/rembg

[5]Adobe Firefly, a text-to-image generation tool: https://firefly.adobe.com

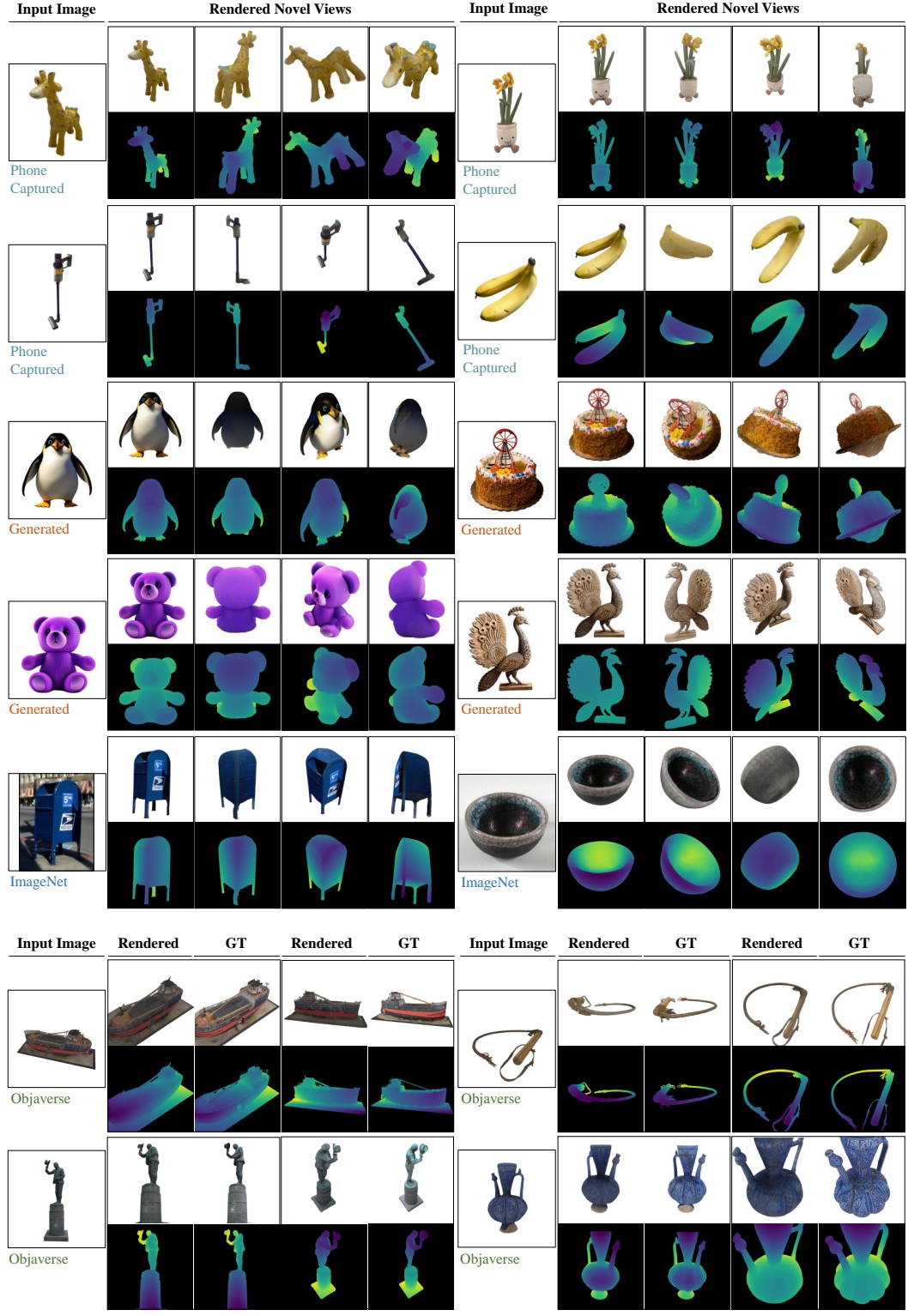

Figure 2: Rendered novel views (RGB and depth) of shapes reconstructed by our LRM from single images. None of the images are observed by the model during training. Generated images are created using Adobe Firefly. The last two rows compare our results to the rendered ground truth images of Objaverse objects (GT). Please zoom in for clearer visualization.

| Input Image | Ours | One-2-3-45 | Input Image | Ours | One-2-3-45 |
|---|---|---|---|---|---|

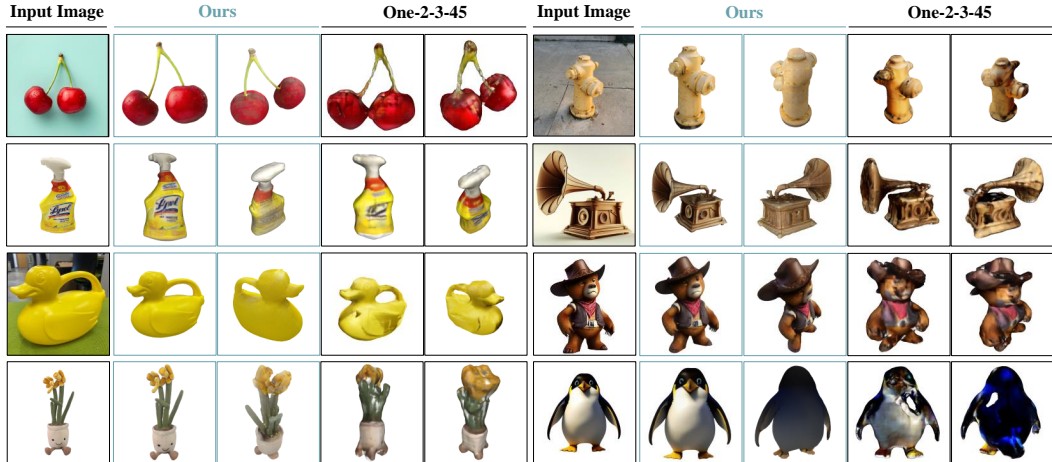

Figure 3: Comparison to One-2-3-45 (Liu et al., 2023a). To avoid cherry-picking, input images in the first three rows are selected from the examples provided in One-2-3-45's paper or demo page. None of the images are observed by our model during training. Please zoom in for clearer visualization.

| Input Image | Rendered Novel Views | Input Image | Rendered Novel Views | Input Image | Rendered Novel Views |
|---|---|---|---|---|---|

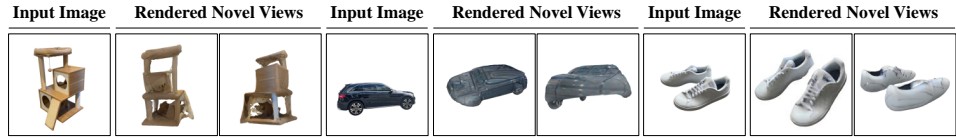

Figure 4: Failure cases of our method. All three examples show blurry textures for occluded regions, and distortion due to the largely inaccurate assumption of the camera parameters.

**Network Architecture** We apply the ViT-B/16 model of pre-trained DINO as the image encoder, which takes $512 \times 512$ RGB images as input and produces 1025 feature tokens (1024 patch-wise features plus one [CLS] features) of dimension 768 ($d_E$) (Caron et al., 2021). The image-to-triplane decoder and the $\text{MLP}^{nerf}$ are of 16 and 10 layers with hidden dimensions 1024 ($d_D$) and 64, respectively. The triplane dimension is 80 ($d_T$). For neural rendering, LRM uniformly samples 128 points for each ray and renders $128 \times 128$ resolution images for supervision. We also use the deferred back-propagation introduced in ARF (Zhang et al., 2022) to save GPU memory.

**Training** We train LRM on 128 NVIDIA (40G) A100 GPUs with batch size 1024 (1024 different shapes per iteration) for 30 epochs, taking about 3 days to complete. Each epoch contains one copy of the rendered image data from Objaverse and three copies of the video frame data from MvImgNet to balance the amount of synthetic and real data. For each sample, we use 3 randomly chosen side views (*i.e.*, the total views $V = 4$) to supervise the shape reconstruction, and we set the coefficient $\lambda = 2.0$ for $\mathcal{L}_{\text{LPIPS}}$. We apply the AdamW optimizer (Loshchilov & Hutter, 2017) and set the learning rate to $4 \times 10^{-4}$ with a cosine schedule (Loshchilov & Hutter, 2016). We numerically analyze the influence of data, training, and model hyper-parameters in the Appendix.

**Inference** During inference, LRM takes an arbitrary image as input (squared and background removed) and assumes the unknown camera parameters to be the normalized cameras that we applied to train the Objaverse data. We query a resolution of $384 \times 384 \times 384$ points from the reconstructed triplane-NeRF and extract the mesh using Marching Cubes (Lorensen & Cline, 1998). This entire process only takes less than 5 seconds to complete on a single NVIDIA A100 GPU.

## 4.3 RESULTS

We visualize the novel views of shapes reconstructed from real, generated, and rendered images from various datasets (Fig. 2), compare our method with a concurrent work (Liu et al., 2023a) (Fig. 3), and summarize some failure cases of our method (Sec. 4.3.2). Numerical comparisons to other methods, and analyses of data, model architecture, and supervision can be found in the Appendix.

### 4.3.1 VISUALIZATION

Figure 2 visualizes some examples of the shapes reconstructed from single images. Overall, the results show very high fidelity for diverse inputs, including real, generated, and rendered images of various subjects with distinct textures. Not only is complex geometry correctly modeled (*e.g.* flower, flagon, and wipe), but also the high-frequency details, such as the texture of the wood peafowl, are preserved, both reflecting the great generalization ability of our model. From the asymmetric examples, giraffe, penguin, and bear, we can see that LRM can infer semantically reasonable occluded portion of the shapes, which implies effective cross-shape priors have been learned.

In Figure 3, we compare LRM with One-2-3-45, a concurrent work to ours that achieves state-of-the-art single image to 3D reconstruction by generating multi-view images with 2D diffusion models (Liu et al., 2023a). To avoid cherry-picking, we directly test our method on the example images provided in their paper or demo page[6]. We can see that our method produces much sharper details and consistent surfaces. In the last row of the figure, we test One-2-3-45 with two examples used in Figure 2, showing much worse reconstruction results.

### 4.3.2 LIMITATIONS

Despite the high-quality single-image-to-3D results we have shown, our method still has a few limitations. First, our LRM tends to produce blurry textures for occluded regions, as shown in Figure 4. We conjecture that this is due to the fact that the single-image-to-3D problem is inherently probabilistic, i.e., multiple plausible solutions exist for the unseen region, but our model is deterministic and is likely producing averaged modes of the unseens. Second, during inference time, we assign a set of fixed camera intrinsics and extrinsics (same as our Objaverse training data) to the test images. These camera parameters may not align well with the ground truth, especially when the images are cropped and resized, causing large changes to Field-of-View (FoV) and principal points. Figure 4 shows that incorrect assumptions of the camera parameters can lead to distorted shape reconstruction. Third, we only address images of objects without background; handling the background (Zhang et al., 2020; Barron et al., 2022), as well as complex scenes, is beyond the scope of this work. Finally, we assume Lambertian objects and omit the view-dependent modelling (Mildenhall et al., 2021) in our predicted NeRF. Therefore, we cannot faithfully reconstruct the view-dependent appearance of some real-world materials, *e.g.*, shiny metals, glossy ceramics, etc.

## 5   CONCLUSION

In this paper, we propose LRM, the first large transformer-based framework to learn an expressive 3D prior from a million 3D data to reconstruct objects from single images. LRM is very efficient in training and inference; it is a fully-differentiable network that can be trained end-to-end with simple image reconstruction losses and only takes five seconds to render a high-fidelity 3D shape, thus enabling a wide range of real-world applications. In the era of large-scale learning, we hope our idea can inspire future research to explore data-driven 3D large reconstruction models that generalize well to arbitrary in-the-wild images.

**Future Directions**   In addition to addressing the limitations mentioned in Sec. 4.3.2, we suggest two future directions of our research; (1) Scaling up the model and training data: with the simplest transformer-based design and minimal regularization, LRM can be easily scaled to a larger and more capable network, including but not limited to applying a larger image encoder, adding more attention layers to the image-to-triplane decoder, and increasing the resolution of triplane representations. On the other hand, LRM only requires multi-view images for supervision, hence a wide range of 3D, video, and image datasets can be exploited in training. We expect both approaches to be promising in improving the model's generalization ability and the quality of reconstruction. (2) Extension to multimodal 3D generative models: LRM model builds a pathway for generating novel 3D shapes from language by leveraging a text-to-image generation model to first create 2D images. But more interestingly, we suggest the learned expressive triplane representations could be applied to directly bridge language descriptions and 3D to enable efficient text-to-3D generation and editing (*e.g.*, via latent diffusion (Rombach et al., 2022)). We will explore these ideas in our future research.

---

[6]One-2-3-45 demo page: https://huggingface.co/spaces/One-2-3-45/One-2-3-45.

## ETHICS STATEMENT

LRM proposed in this paper is a deterministic model in which, given the same image as input, the model will infer the identical 3D shape. Unlike generative models that can be used to easily synthesize various undesirable contents (*e.g.*, from language inputs), LRM requests the specific 2D content to exist in the first place. LRM is trained on Objaverse (Deitke et al., 2023) and MvImgNet (Yu et al., 2023) data, which mostly contain ethical content. However, given an unethical or misleading image, LRM could produce unethical 3D objects or 3D disinformation that may be more convincing than the 2D input images (although the reconstructed objects are less realistic than real-world objects).

Image-to-3D reconstruction models like LRM hold the potential to automate tasks currently performed by 3D designers. However, it's worth noting that these tools also have the capacity to foster growth and enhance accessibility within the creative industry.

## REPRODUCIBILITY STATEMENT

Our LRM is built by integrating the publicly available codebases of threestudio[7] (Guo et al., 2023), x-transformers[8], and DINO[9] (Caron et al., 2021), and the model is trained using publicly available data from Objaverse (Deitke et al., 2023) and MvImgNet (Yu et al., 2023). We include very comprehensive data pre-processing, network architecture, and training details in this paper, which greatly facilitate reproducing our LRM.

## ACKNOWLEDGMENT

We want to thank Nathan Carr, Scott Cohen, Hailin Jin, Aseem Agarwala, Tong Sun for their support, and thank Duygu Ceylan, Zexiang Xu, Paul Guerrero, Chun-Hao Huang, Niloy Mitra, Radomir Mech, Vova Kim, Thibault Groueix for constructive feedback on this project. Hao wants to thank Xin for the inspiration as he ran on this road. Yicong wants to thank Prof. Stephen Gould and Ms. Ziwei Wang for their great advice.

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

APPENDICES

# A  BACKGROUND OF MODEL COMPONENTS

## A.1  NeRF

We adopt NeRF (Mildenhall et al., 2021), specifically the compact triplane NeRF variant (Chan et al., 2022), as our 3D representation to predict in LRM. NeRF, when coupled with differentiable volume rendering, can be optimized with just image reconstruction losses.

At the core of NeRF (Mildenhall et al., 2021) and its variants (Chan et al., 2022; Chen et al., 2022a; Müller et al., 2022; Sun et al., 2022) is a spatially-varying color (modeling appearance) and density (modeling geometry) field function. [10] Given a 3D point $\mathbf{p}$, the color and density field $(\mathbf{u}, \sigma)$ can be written as:

$$(\mathbf{u}, \sigma) = \text{MLP}^{nerf}(f_\theta(\mathbf{p})), \tag{7}$$

where the spatial encoding $f_\theta$ is used to facilitate the $\text{MLP}^{nerf}$ to learn high-frequency signals. Different NeRF variants (Chan et al., 2022; Chen et al., 2022a; Müller et al., 2022; Sun et al., 2022) typically differ from each other in terms of the choice of the spatial encoding and the size of the MLP. In this work, we use the triplane spatial encoding function proposed by EG3D (Chan et al., 2022), because of its low tokenization complexity ($O(N^2)$ as opposed to a voxel grid's $O(N^3)$ complexity, where $N$ is spatial resolution).

Images are rendered from NeRF using volume rendering that's trivially differentiable. In detail, for each pixel to render, we cast a ray $\mathbf{r}$ through a NeRF, and use finite point samples $\mathbf{p}_i$ along the ray to compute the volume rendering integral to get the rendered color $\mathbf{u}(\mathbf{r})$:

$$\mathbf{u}(\mathbf{r}) = \sum_i T_i(1 - \exp(-\sigma_i \delta_i))\mathbf{u}_i, \tag{8}$$

$$T_i = \exp(-\sum_{j=1}^{i-1} \sigma_j \delta_j), \tag{9}$$

where $(\mathbf{u}_i, \sigma_i) = \text{MLP}_\phi(f_\theta(\mathbf{p}_i))$ and $\delta_i$ is the distance between point $\mathbf{p}_i$ and $\mathbf{p}_{i+1}$.

## A.2  TRANSFORMER LAYERS

In this subsection, we provide the details of the layers used in the transformer decoder (Vaswani et al., 2017) as a background. For the Vision Transformer encoder, please refer to the original DINO paper (Caron et al., 2021) for implementation details.

**Attention operator**  Attention operator is an expressive neural operator which converts an input feature $x$ with condition to a sequence of other features $\{y_i\}$. It first computes the attention score $\alpha_i$ by using the dot product between the input $x$ and each condition feature $y_i$. An additional $\text{softmax}$ is added after the dot products to normalize the weights to a summation of 1. This attention score measures the relationship between input and conditions. Then the output is the weighted summation of the conditions $\{y_i\}$ with respect to the attention score $\alpha_i$.

$$\alpha_i = \text{softmax}_i\{x^\top y_i\} \tag{10}$$

$$\text{Attn}(x; \{y_i\}_i) = \sum_i \alpha_i y_i \tag{11}$$

For some specific cases (*e.g.*, in the transformer attention layer below), the attention operator wants to differentiate the vectors used in calculating the attention score and the vectors for final outputs. Thus it will introduce another set of 'value' vectors $\{z_i\}_i$, and treat the $\{y_i\}_i$ as corresponding 'key' vectors. Taking this into consideration, the formula would become

$$\alpha_i = \text{softmax}_i\{x^\top y_i\} \tag{12}$$

$$\text{Attn}(x; \{y_i\}_i, \{z_i\}_i) = \sum_i \alpha_i z_i \tag{13}$$

---

[10]To simplify the discussion, we ignore the view-dependent modeling in NeRF (Mildenhall et al., 2021).

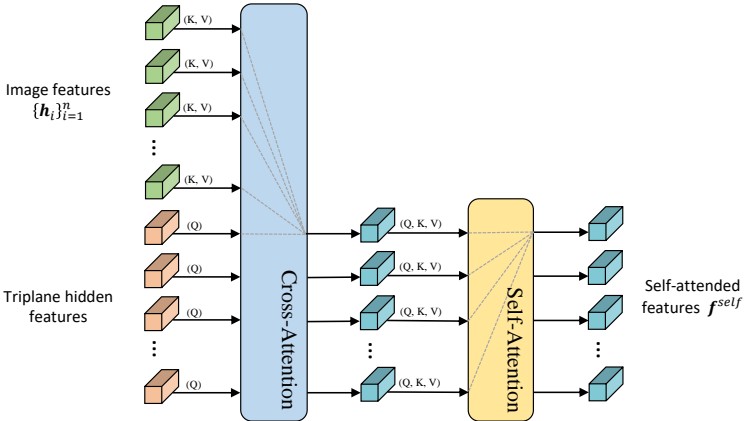

Figure 5: Visual illustration of the cross-attention and self-attention in LRM's image-to-triplane decoder.

**Multi-head Attention** The attention operator described above only attends to the condition features once to get the attention vector. However, the actual attention might contain multiple modes. Thus, the multi-head attention (Vaswani et al., 2017) is proposed. The multi-head attention is implemented by first splitting the input features into smaller queries.

$$[x^1, \ldots, x^{nh}] = x \tag{14}$$

where $nh$ is the number of heads. Meanwhile, $y_i$ and $z_i$ are split into $\{y_i^k\}_k$ and $\{z_i^k\}_k$ in a similar way. After that, the output of each head is computed independently and the final output is a concatenation of heads' outputs.

$$out^k = \text{Attn}(x^k; \{y_i^k\}_i, \{z_i^k\}_i) \tag{15}$$
$$\text{MultiHeadAttn}(x; \{y_i\}_i, \{z_i\}_i) = [out^1, \ldots, out^{nh}] \tag{16}$$

**Attention Layers in Transformer** The detailed attention layers in transformer utilize the above multi-head attention with more linear layers. Here are the formulas for the self-attention layer (see the right yellow 'Self-Attention' block in Fig. 5). The layer first projects the input feature sequence $f = \{f_j\}_j$ to query $q$, key $k$, and value $v$ vectors with linear layers. Then the multi-head attention is applied. There is one more linear layer over the output. We also follow the recent papers (Chowdhery et al., 2022; Touvron et al., 2023) to remove the bias terms in the attention layers.

$$q_j = W_q f_j \tag{17}$$
$$k_i = W_k f_i \tag{18}$$
$$v_i = W_v f_i \tag{19}$$
$$o_j = \text{MultiHeadAttn}(q_j; \{k_i\}_i, \{v_i\}_i) \tag{20}$$
$$\text{SelfAttn}(f_j; \{f_j\}_j) = W_{\text{out}} o_j \tag{21}$$
$$\tag{22}$$

The cross-attention layer is defined similarly (see the left blue 'Cross-Attention' block in Fig. 5). The only difference to the self-attention layer is that the $W_k$ and $W_v$ is applied to the condition vectors (*e.g.*, the image features $h$ in our example).

**MLP layers in Transformer** The Transformer model architecture applies the MLP layer (multi-layer perceptron) to do channel mixing (*i.e.*, mix the information from different feature dimensions). We follow the original transformer paper (Vaswani et al., 2017) for the implementation. The MLP layer contains two linear layers with a GELU (Hendrycks & Gimpel, 2023) activation in between. The intermediate hidden dimension is 4 times of the model dimension.

**Layer Normalization**   We take the default LayerNorm (LN) implementation in PyTorch (Paszke et al., 2019). Besides the LN layers in ModLN as in Sec. 3.2, we follow the Pre-LN architecture to also apply LN to the final output of transformers, *e.g.*, the output of ViT and also the output of transformer decoder.

**Positional Encoding**   The positional embedding in ViT (Dosovitskiy et al., 2020) is bilinearly up-sampled from its original resolution ($14 \times 14$ for input $224 \times 224$) to match our higher input resolution ($32 \times 32$ for input $512 \times 512$).

## B   TRAINING SETUP

We specify the training setup of our LRM. Apart from the information that we provided in Sec. 4.2, we apply a cosine schedule (Loshchilov & Hutter, 2016) with 3000 warm-up iterations. We set the second beta parameter ($\beta_2$) of the AdamW optimizer (Loshchilov & Hutter, 2017) to be 0.95. We apply a gradient clipping of 1.0 and a weight decay of 0.05. The weight decay are only applied on the weights that are not bias and not in the layer normalization layer. We use BF16 precision in in the mixed precision training. To save computational cost in training, we resize the reference novel views from $512 \times 512$ to a randomly chosen resolution between $128 \times 128$ and $384 \times 384$ and only ask the model to reconstruct a randomly selected $128 \times 128$ region. With this design, we can possibly increase the effective resolution of the model.

## C   COMPARISON WITH SoTA

We provide a quantitative comparison to the stat-of-the-art methods Point-E (Nichol et al., 2022), Shap-E (Jun & Nichol, 2023), and One-2-3-45 (Liu et al., 2023a). Point-E trains an image-to-3D point cloud diffusion model, Shap-E encodes point clouds to latent representations and trains a diffusion model on the latents to generate parameters of a 3D implicit function, and One-2-3-45 reconstructs multi-view images generated with a 2D diffusion model. We randomly selected 100 objects from the Google Scanned Objects (GSO) dataset (Downs et al., 2022) and measured the novel view synthetic quality of 20 reference views (FID, CLIP-Similarity (Radford et al., 2021), PSNR, LPIPS (Zhang et al., 2018)) and the geometric quality (Chamfer Distance), as shown in the Table below. We can see that our LRM consistently outperforms previous approaches in all metrics.

Table 1: Comparison between LRM and state-of-the-art 3D generative models on Google Scanned Objects dataset (100 randomly selected objects and 20 reference views).

| Models | GSO Evaluation | | | | |
|---|---|---|---|---|---|
| | FID↓ | CLIP-Similarity↑ | PSNR↑ | LPIPS↓ | Chamfer Distance↓ |
| Point-E | 123.70 | 0.741 | 15.60 | 0.308 | 0.099 |
| Shap-E | 97.05 | 0.805 | 14.36 | 0.289 | 0.085 |
| One-2-3-45 | 139.24 | 0.713 | 12.42 | 0.448 | 0.123 |
| LRM (ours) | **31.44** | **0.902** | **19.60** | **0.163** | **0.053** |

We would like to discuss further the difference between LRM and the large-scale approaches Point-E and Shap-E. The models of Point-E and Shap-E contain hundreds of millions of learnable parameters and are trained with several million 3D assets (unknown data source and unknown computational cost from their papers). In terms of the network and dataset sizes, our LRM has 500 million learnable parameters, and it is trained on 1 million 3D data (publicly accessible), which does not show an advantage. In terms of the network architecture, Point-E, Shap-E, and LRM all use transformer-based models and apply cross-attention for inter-modality modeling (*i.e.*, image-to-point cloud, point cloud+image-to-3D latents, and image-to-triplane, respectively). We hypothesize it is the choice of very compact and expressive triplane representation together with an end-to-end trainable framework that enables the effective scaling of LRM and its adequate learning on large datasets (Objaverse and MvImgNet). Compared to the unstructured point cloud representation applied in Point-E and Shap-E, LRM applies the structured triplane representation that is aligned with the world frame, which naturally facilitates 2D-to-3D projection. It is also worth mentioning that Point-E uses 4K points (as tokens) and Shap-E uses 16K points (as tokens), but our LRM only uses

$3{\times}32{\times}32{=}3072$ triplane tokens, which largely reduce the modeling complexity. Additionally, compared to the two-stage approach in Shape-E, which attempts to generate latents that can produce the parameters of implicit 3D functions through a diffusion model, our LRM directly maps 2D images to triplanes, which should be much more stable and efficient to learn. Overall, we suggest that LRM is a more data-friendly and efficient model than Point-E and Shap-E.

# D  ANALYSES

We evaluate the effect of data, model hyper-parameters, and training methods on the performance of LRM, measuring by PSNR, CLIP-Similarity (Radford et al., 2021), SSIM (Wang et al., 2004) and LPIPS (Zhang et al., 2018) of the rendered novel views. Note that due to the large training cost of our final model, the following analytic experiments use a much smaller version of LRM model as the baseline (indicated by orange shaded rows in the tables). Specifically, we scale down the image-to-triplane decoder to 12 cross-attention layers, change the input image resolution to 256, triplane latent dimension to 32, rendering resolution in training to 64, and use 96 samples per ray for rendering $64{\times}64$ images for supervision. We only train each model on 32 NVIDIA A100 GPUs for 15 epochs, and the resulting difference can be seen in Table 2. We are aware that some observations might change if we scale up the model, but most of the conclusions should be general and consistent.

Table 2: Comparison between the final model and the baseline for analysis.

| Models | Unseen Evaluation | | | |
| --- | --- | --- | --- | --- |
| | PSNR↑ | CLIP-Similarity↑ | SSIM↑ | LPIPS↓ |
| Final | **20.1** | **91.0** | **79.7** | **16.0** |
| Baseline | 19.0 | 87.8 | 77.4 | 19.1 |

## D.1  SYNTHETIC VS. REAL DATA

Table 3 compares the influence of using synthetic 3D data from the Objaverse (Deitke et al., 2023) and real video data from the MvImgNet (Yu et al., 2023) in training. Results show that removing real data causes an obvious drop for all the metrics, despite the fact our synthetic 3D dataset contains $3{\times}$ more shapes than MvImgNet. One potential reason is that the real data have much more variation in the lighting, the size of the target, and the camera poses, which effectively benefits the learning. Future work could augment the rendering of synthetic shapes to adequately utilize those abundant data. Nevertheless, combining the two datasets leads to substantially better results than training on any one of them alone.

Table 3: Influence of training datasets.

| Data | Unseen Evaluation | | | |
| --- | --- | --- | --- | --- |
| | PSNR↑ | CLIP-Similarity↑ | SSIM↑ | LPIPS↓ |
| Synthetic (Objaverse) | 15.5 | 84.7 | 70.3 | 29.3 |
| Real (MvImgNet) | 17.5 | 85.7 | 75.7 | 22.0 |
| Synthetic+Real | **19.0** | **87.8** | **77.4** | **19.1** |

## D.2  NUMBER OF VIEWS IN TRAINING DATA

In Table 4, we conduct experiments with all data but limit the number of training views per shape. For example, for *Train Views=8*, we use only a random subset of 8 views per shape and keep randomly sampling 4 views from the above subset at each training step. The results show that more views can lead to better results, possibly because of more diverse data. While the growth is saturated at 16 views, adding more views does not lead to worse results.

## D.3  MODEL HYPER-PARAMETERS

Table 5 presents the results of having a different number of cross-attention layers in the image-to-triplane decoder. There is a slight trend indicating that the scores can be improved by having a

Table 4: Effect of the number of different views per shape in training. 32+ indicates some video data in MvImgNet contain more than 32 views per shape, which we apply all of them in training.

| Train Views | Unseen Evaluation | | | |
|---|---|---|---|---|
| | PSNR↑ | CLIP-Similarity↑ | SSIM↑ | LPIPS↓ |
| 4 | 18.8 | 86.7 | 77.5 | 19.8 |
| 8 | 18.9 | 87.3 | 77.5 | 19.4 |
| 16 | **19.1** | **87.9** | **77.6** | **19.0** |
| 32+ | 19.0 | 87.8 | 77.4 | 19.1 |

deeper model, especially for the latent semantic and perceptual similarity measurements CLIP and LPIPS, implying that the network models better representations for reconstructing higher-quality images.

We also evaluate the influence of the number of MLP layers in NeRF (Table 6). Results show that it is unnecessary to have a very large network, and there seems to be a sweet spot around two to four layers. This observation is consistent with EG3D (Chan et al., 2022) where the information of shapes is encoded by the triplane and such MLP is only a shallow model for projecting triplane features to color and density.

As shown in Table 7, we found that increasing the triplane resolution leads to better image quality. Note that, in this experiment, we only use a deconvolution layer to upsample the $32\times32\times32$ triplane produced by LRM's decoder, whereas we suspect a large improvement could be seen by increasing the quantity of input spatial-positional embeddings to query more fine-grained image details. However, such an approach will dramatically increase the computational cost, we leave this exploration to future research.

Table 5: Effect of the number of cross-attention layers in image-to-triplane decoder.

| CrossAttn Layers | Unseen Evaluation | | | |
|---|---|---|---|---|
| | PSNR↑ | CLIP-Similarity↑ | SSIM↑ | LPIPS↓ |
| 6 | 19.0 | 87.7 | **77.6** | 19.1 |
| 16 | 19.0 | 87.8 | 77.4 | 19.1 |
| 24 | **19.1** | **88.0** | **77.6** | **18.9** |

Table 6: Effect of the number of MLP layers in NeRF.

| NeRF MLP Layers | Unseen Evaluation | | | |
|---|---|---|---|---|
| | PSNR↑ | CLIP-Similarity↑ | SSIM↑ | LPIPS↓ |
| 2 | **19.2** | 87.7 | **77.8** | **18.9** |
| 6 | 19.1 | **88.0** | 77.6 | 19.0 |
| 12 | 19.0 | 87.8 | 77.4 | 19.1 |
| 14 | 19.1 | 87.2 | 77.6 | 19.0 |

Table 7: Effect of the resolution of triplane. For *64up* and *128up*, we apply additional $2\times2$ and $4\times4$ deconvolution layers, respectively, to upsample a *Res.* 32 triplane.

| Triplane Res. | Unseen Evaluation | | | |
|---|---|---|---|---|
| | PSNR↑ | CLIP-Similarity↑ | SSIM↑ | LPIPS↓ |
| 32 | 18.9 | 86.3 | 77.2 | 19.7 |
| 64up | **19.0** | 87.8 | 77.4 | 19.1 |
| 128up | **19.0** | **88.3** | **77.5** | **19.0** |

## D.4 CAMERA POSE

As we have discussed in the Main Paper, normalizing camera poses in training has a huge impact on the generalization of input views. We can see from Table 8 that when no modification is applied (*None*), LRM produces the worst results. Augmenting camera poses with a *Random* rotation

greatly improves the results since the model learns a more general image-to-triplane projection via decoupled views and camera poses. However, such unconstrained projection is very difficult to learn. We therefore *Normalized* all camera poses so that all images are projected onto the triplane from the same direction, allowing the model to adequately learn and utilize the cross-shape prior for reconstruction.

Table 8: Effect of camera pose normalization.

| Camera Pose | Unseen Evaluation | | | |
| | PSNR↑ | CLIP-Similarity↑ | SSIM↑ | LPIPS↓ |
|---|---|---|---|---|
| None | 15.3 | 83.4 | 70.1 | 28.9 |
| Random | 18.0 | 85.6 | 75.7 | 21.1 |
| Normalized | **19.0** | **87.8** | **77.4** | **19.1** |

### D.5   IMAGE QUANTITY AND RESOLUTION

Table 9 and Table 10 study the influence of the number of side views supervision for each sample and the effect of image rendering resolution in training. Results indicate that as the quantity of side views increases, the reconstructed image quality improves. Having more views allows the model to better correlate the appearance and geometry of different parts of the same shape, and facilitates inferring multi-view consistent results. Moreover, using a higher rendering resolution of images in training largely improves the results, as the model is encouraged to learn more high-frequency details.

Table 9: Influence of the number of side views applied for each training sample.

| Side Views | Unseen Evaluation | | | |
| | PSNR↑ | CLIP-Similarity↑ | SSIM↑ | LPIPS↓ |
|---|---|---|---|---|
| 1 | 18.7 | 87.7 | 77.2 | 19.7 |
| 2 | 18.7 | 87.5 | 77.2 | 19.6 |
| 3 | 19.0 | **87.8** | 77.4 | 19.1 |
| 4 | **19.1** | **87.8** | **77.6** | **18.9** |

Table 10: Influence of the rendering resolution of images in training.

| Render Res. | Unseen Evaluation | | | |
| | PSNR↑ | CLIP-Similarity↑ | SSIM↑ | LPIPS↓ |
|---|---|---|---|---|
| 32 | 18.8 | 86.3 | 77.0 | 20.1 |
| 64 | 19.0 | 87.8 | 77.4 | 19.1 |
| 128 | **19.4** | **89.0** | **78.3** | **18.0** |

### D.6   LPIPS LOSS

Lastly, we found that our LPIPS objective (Zhang et al., 2018) has a huge impact on the results. Removing it from training will decrease the CLIP-Similarity, SSIM, and LPIPS scores to 74.7, 76.4, and 29.4, respectively.

## E   VISUALIZATIONS

We present more visualizations of the reconstructed 3D shapes in the following pages. The input images include photos captured by our phone camera, images from Objaverse (Deitke et al., 2023), MvImgNet (Yu et al., 2023), ImageNet (Deng et al., 2009), Google Scanned Objects (Downs et al., 2022), Amazon Berkeley Objects (Collins et al., 2022), and images generated by the Adobe Firefly[11]. We implement a heuristic function to pre-process the camera-captured images, generated

---

[11]Adobe Firefly, a text-to-image generation tool: https://firefly.adobe.com.

images, and images from MvImgNet and ImageNet. The function removes the image background with an off-the-shelf package[12], followed by cropping out the target object, rescaling the target to a suitable size and centering the target on a square white figure. All input images are never seen by the model in training. Please visit our project webpage https://yiconghong.me/LRM/ for video demonstrations and interactable 3D meshes.

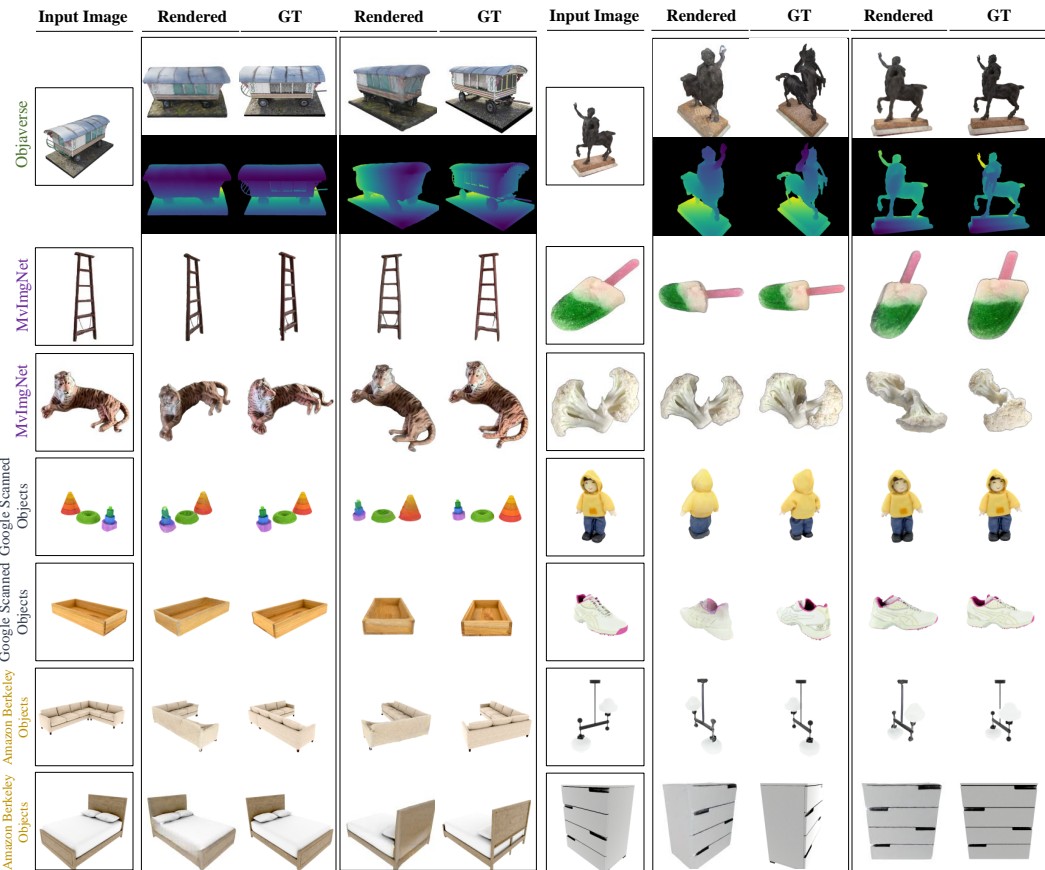

Figure 6: Comparison between LRM rendered novel views and the ground truth images (GT). None of the images are observed by the model during training. The GT depth images of Objaverse are rendered from the 3D models. Please zoom in for clearer visualization.

---

[12]Rembg package, a tool to remove image background: https://pypi.org/project/rembg

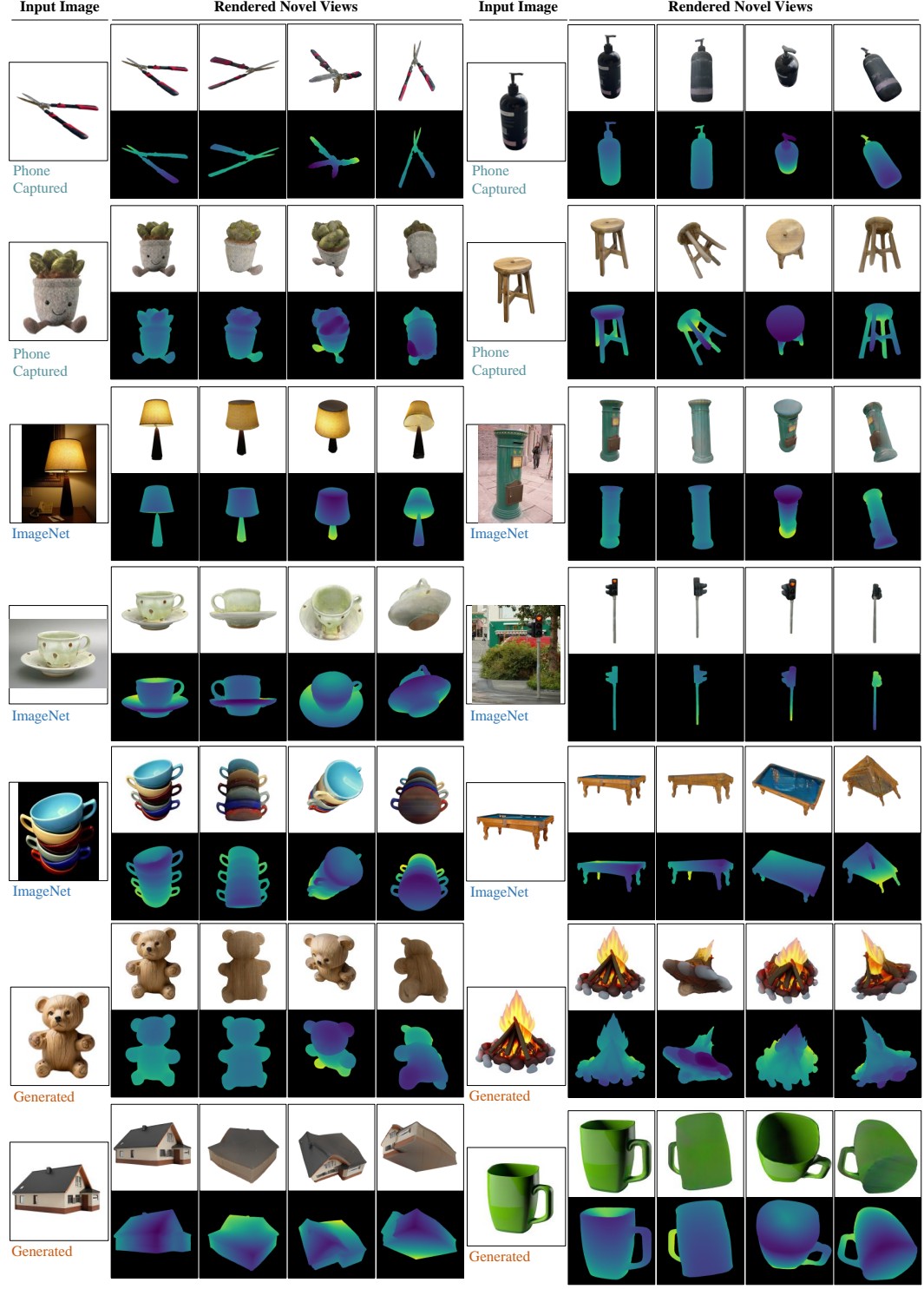

Figure 7: Rendered novel views (RGB and Depth) of shapes reconstructed by our LRM from single images. None of the images are observed by the model during training. Generated images are created by the Adobe Firefly. Please zoom in for clearer visualization.

