# OpenReview forum: "LRM: Large Reconstruction Model for Single Image to 3D"
_ICLR.cc/2024/Conference — ICLR 2024 oral_

### Official Review · Reviewer_eyyQ · 2023-10-22

**Soundness:** 4 excellent
**Presentation:** 4 excellent
**Contribution:** 4 excellent
**Rating:** 10
**Confidence:** 5

**Summary:**

The paper proposes a large reconstruction model LRM, predicts the 3D model of an object from a single input image within 5 seconds .LRM adopts a transformer-based architecture with 500 million learnable parameters to directly predict a neural radiance field from the input image The model is trained with multi-view data containing around 1 million objects, including both synthetic renderings from Objaverse and real captures from MVImgNet. This make LRM to be highly generalizable and produce high-quality 3D reconstructions from various testing inputs including real-world in-the-wild captures and images from generative models.

**Strengths:**

The strengths of the paper are as follows:

1. The paper designs a 3D foundation model for reconstruction, which intrinsically establishes a 3D prior knowledge model. It fills the gap between 2D and 3D foundation generative model. This contribution makes this work not only achieve SOTA 3D generation result, but advances the 3D deep learning. It has potential to influence the 3D learning community as BERT for NLP, or DALLE2/ Muse for 2D generation.

2. The paper figures out a learnable positional encoding to generalize NeRF generation conditioning on 2d image features. The architecture pave the way for other 3D generative models to scale up.

3. The paper drastically improves the generation speed for 3D assets, which will help democratize the technologies in the burgeoning fields like 2d-to-3d or text-to-3d. The potential quasi-real time applications can make 3D content generation more accessible to common users, which in turns, will benefit the development of the community.

**Weaknesses:**

1. As mentioned in the limitation, the back side of the model (e.g., figure 3 penguin) is blurry and besides that, the color tone/ saturation seems a bit inconsistent with the frontal view.

2. The model relies on the correct camera parameters and normalization of camera, which still based on the fact that the objects in the training dataset (objectaverse?) have a certain distribution of orientation. The more general way to formulate the scene is to assume the world coordinate is the camera coordinate (extrinsic = np.eye(4)),  which make the object always face forward to the camera. Although, from the geometric learning works in shapenet, this approach is more challenging than appropriating the object orientation.

3. The approach relies on multiview data which limits the model incorporating more diversity in data source of single view images.

**Questions:**

The biggest question is whether the author plan to release the code and training setting, so the community can quickly adopts the direction of generative model for direct 3D generation, in stead of using SDS or nerf2nerf which learns nothing of 3D prior.

In addition, the most comparable models of large-scale direct 3D generation are Point-E and Shap-E. From the example shown in the paper, it seems LRM performs way better than both and more analysis of the better quality is preferred. Is it because LRM use objaverse and MVImage net, both contain much more items than what Point-E and Shap-E use or is it because of the design of learnable positional encoding and other architectures of LRM? More analysis or explanation can further elevate this paper to another level.

---

> ### Author Response · Authors · 2023-11-16
> **Respond to Reviewer 4 (eyyQ) -- Part 1**
>
> We thank Reviewer 4 (eyyQ) for acknowledging the contribution of our paper and providing very constructive feedback. Please see our response to the comments below.
>
> ### **1. Quality of the Back Side (Respond to R4W1)**
> We thank the reviewer for pointing out this limitation. As discussed in Section 4.3.2, multiple plausible solutions exist for the occluded side, but our model is deterministic and likely produces averaged modes of the unseen, which could cause blurry textures and shapes.
> This issue is a joint effect of LRM's deterministic approach and an L2 loss and applying MVImgNet data for training, which mostly only covers 180 degrees of view. A similar issue can be seen in the Masked Autoencoders [1], where the model tends to reconstruct blurry contents when a large block of region is removed from an image. One potential solution is to post-optimize the LRM's output (e.g., using generative prior and SDS loss proposed by DreamFusion [2]) at the cost of extra processing time.
>
> On the other hand, our ongoing research found that LRM can be easily extended to accept sparse multi-view inputs by passing encoded multi-view image features to the image-to-triplane decoder, and making the positional embeddings query from features of all images via cross-attention to construct the triplane. This model can create very high-fidelity shapes where the issue of blurry occluded portions can be largely eliminated due to the higher coverage of the 3D object. We will make the multiview LRM paper public to the research community.
>
> [1] Masked Autoencoders Are Scalable Vision Learners. He et al., 2021.
>
> [2] DreamFusion: Text-to-3D using 2D Diffusion. Poole et al., 2022.
>
>
> ### **2.Object Orientation and Camera Coordinate (Respond to R4W2)**
> We thank the reviewer for this comment, but we are not entirely sure that we fully understand the question. We will try to respond. If we missed anything, please elaborate for us. Thank you!
>
> Our model relies on the correct camera parameters in training, and we normalize the cameras in training. We use 3D data in Objaverse (3D models) and MvImgNet (videos of objects) for training; the majority of objects in Objaverse have a certain orientation, but objects in MvImgNet do not. As specified in Section 4.1, to render multiviews from objects in Objaverse, we normalize the 3D shape to the
> box $[-1,1]^{3}$ in world space and render 32 random views with the same camera pointing toward the world center at arbitrary poses. The camera poses are sampled from a ball of radius $[1.5, 3.0]$ and with height in range $[0.75, 1.60]$.
>
> During training, we normalize the camera before feeding an image to LRM (details in Section 4.2), resulting in a very similar formulation as the \textit{general way} described by the Reviewer. Specifically, regardless of the initial position of the camera, we normalize its poses to position $[0,-2,0]$, with the camera's vertical axis aligned with the upward z-axis in the world frame (analogy to the case of extrinsic=[[1,0,0,0],[0,1,0,-2],[0,0,1,0],[0,0,0,1]] mentioned by the Reviewer). Such normalization does not alter the input image, so it actually rotates and scales the object accordingly. This means that regardless of where the input image is taken from, it will be projected onto the triplane from exactly the same direction. As discussed in Section 3.2, this method greatly reduces the optimization space of triplane features and facilitates model convergence.
>
> Finally, at inference, LRM assumes the unknown camera parameters to be the normalized cameras that we applied to train the Objaverse data (Section 4.2).
>
>
> ### **3. Training Requires Multiview Data (Respond to R4W3)**
>  The training of LRM requires multiview data; however, we have witnessed continuous growth in the size and diversity of 3D datasets (e.g., the latest Objaverse-XL dataset contains 10.2 million 3D assets), and there are enormous amounts of videos that can potentially provide multiview supervision for reconstruction. Although the number of single-view images is far larger, we remain positive about the amount of multiview data for large-scale 3D learning in the future.
> On the other hand, the recent approach 3DGP [2], which is trained on single-view images, still struggles to produce consistent and complete shapes.
> We believe that incorporating single-view data and limited multiview data is an open and highly valuable research question, which we will leave to future work.
>
> [1] Objaverse-XL: A Universe of 10M+ 3D Objects. Deitke et al., 2023.
>
> [2] 3D Generation on ImageNet. Skorokhodov et al., 2023.

---

> ### Author Response · Authors · 2023-11-16
> **Respond to Reviewer 4 (eyyQ) -- Part 2**
>
> ### **4. Release Code and Training Setting (Respond to R4Q1)**
> We thank the reviewer for raising this important question. The release plan is still under discussion, and the authors are waiting for more guidance on this. Meanwhile, we have provided very comprehensive model architecture and training details in the paper (Section 3 and Section 4) and Appendix (C and D) for reproducing LRM. In addition to the final model that we presented in the paper, we also attempted to train a smaller version of LRM.
> The model uses the same architecture as the baseline model we showed in Appendix C Table 1, and we trained it on 8 NVIDIA (40G) A100 GPUs for 4 days with batch size 64 and gradient accumulation of 4 steps. This model achieves performance between the baseline and the final models shown in Table 1, which is able to produce reasonable results for research purposes.
>
>
> ### **5. Comparison to Point-E and Shap-E (Respond to R4Q2)**
> We thank the reviewer for this very constructive suggestion of an in-depth comparison with Point-E [1] and Shap-E [2]. We mentioned Point-E and Shap-E in our paper, and we will discuss more here. Point-E trains an image-to-3D point cloud diffusion model, and Shap-E encodes point clouds to latent representations and trains a diffusion model on the latents to generate parameters of a 3D implicit function. Both models contain hundreds of millions of learnable parameters and are trained with several million 3D assets (unknown data source and unknown computational cost).
> In terms of the network and dataset sizes, our LRM has 500 million learnable parameters, and it is trained on 1 million 3D data, which does not show an advantage. In terms of the network architecture, Point-E, Shap-E, and LRM all use transformer-based models and apply cross-attention for inter-modality modeling (i.e., image-to-point cloud, point cloud+image-to-3D latents, and image-to-triplane).
>
> We hypothesize it is the choice of very compact and expressive triplane representation together with an end-to-end trainable framework that enables the scaling of the LRM and its adequate learning on large datasets (Objaverse and MvImgNet).
> Compared to the unstructured point cloud representation applied in Point-E and Shap-E, LRM applies the structured triplane representation that is aligned with the world frame, which naturally facilitates 2D-to-3D projection. It is also worth mentioning that Point-E uses 4K points (as tokens) and Shap-E uses 16K points (as tokens), but our LRM only uses $3{\times}32{\times}32{=}3072$ triplane tokens, which largely reduce the modeling complexity. Additionally, compared to the two-stage approach in Shape-E, which attempts to generate latents that can produce the parameters of implicit 3D functions through a diffusion model, our LRM directly maps 2D images to triplanes, which should be much more stable and efficient to learn.
>
> Overall, we suggest that LRM is a more data-friendly and efficient model than Point-E and Shap-E. We will further extend this discussion and add it to our paper.
>
> [1] Point-E: A System for Generating 3D Point Clouds from Complex Prompts. Nichol et al., 2022.
>
> [2] Shap-E: Generating Conditional 3D Implicit
> Functions. Jun and Nichol, 2023.

---

> ### Comment · Reviewer_eyyQ · 2023-11-20
> **Score adjustment**
>
> I'm satisfied with the response and appreciate the in-depth analysis from the authors. I therefore i raised my score to strong accept. As a part of the ICLR community not only like to see papers that are theoretically sound, but actually work well.

---

### Official Review · Reviewer_ZEtF · 2023-10-26

**Soundness:** 4 excellent
**Presentation:** 3 good
**Contribution:** 4 excellent
**Rating:** 8
**Confidence:** 4

**Summary:**

This paper proposes the Large Reconstruction Model (LRM) for image-to-3D task. Different from previous works that distill an image-to-image model for this task, the LRM directly predict the 3D NeRF representation from the input image. In detail, authors utilize a transformer encoder-decoder architecture to predict the triplane NeRF, and trained on a large-scale multi-view dataset including Objaverse and MVImgNet. Qualitative results on images from different source show the superior performance of the proposed LRM. In addition authors also provide an extensive ablation on different design components.

**Strengths:**

This paper is well-written and the results are promising. Although there were papers trying to train a generalizable nerf predictor, this paper proves the possibility of training on a large-scale dataset for the generalizable nerf prediction. To my knowledge, this is the first attempt to train it on scale like Objaverse + MVImgNet. The experiment part is well-organized and sufficient, provide a thorough ablation for different model components.

**Weaknesses:**

I don't think there exists any apparent weakness in the paper. Please refer to the following questions part for my other questions regarding the details of paper.

**Questions:**

1. From a fundamental perspective, this LRM model actually turns the image-to-3D into a deterministic predicting task, which alleviates the ill-pose nature of the task. Does the author think the results are just "fitting" to the training distribution rather than actually predicting / learning a 3D distribution.

2. I think in Equation(4) the index j' actually refers to the set including j? This equation needs to be revised.

3. In supplementary, some implementation details are missing, for example the number of heads in multi-head attention. In addition I don't clearly understand how does the attention behave between image feature [32x32, 768] and embedding [32x32x3, 1024]? I know when mapping to q,k,v the feature dimension can be changed, but here the token numbers are different. In my understanding it results in weight shape [32x32, 32x32x3] and when multiplied by value, results are in [32x32, 1024] shape. But the next layer requires input as [32x32x3, 1024]?

4. In volume rendering, is the 128 points uniformaly sampled or a two-stage sampling strategy or other sampling methods is adopted?

---

> ### Author Response · Authors · 2023-11-16
> **Respond to Reviewer 3 (ZEtF)**
>
> We thank Reviewer 3 (ZEtF) for acknowledging the contribution of our paper and raising valuable questions. Please see our response to the comments below.
>
> ### **1. Fitting vs. Predicting Distribution (Respond to R3Q1)**
> LRM is a deterministic model that learns the distribution of the 3D data in training and fits the given reference image to the learned distribution at inference. Although single-image-to-3D is a one-to-many mapping problem, we made the simplified assumption of one-to-one mapping in this work, which we have shown to be useful in many cases (as visualized in Figure 2, Figure 6, and Figure 7). We would also like to mention that our LRM bridges 2D and 3D representations that can potentially facilitate one-image-to-many-3D-shapes generation in the future. For instance, training latent diffusion models on triplanes predicted by LRM to generate new shapes or reconstruct 3D models from text-to-multiview generations.
>
>
> ### **2. Equation (4) (Respond to R3Q2)**
> Equation (4) expresses a self-attention with residual, where the two input terms are identical sequences of tokens. We use the notation of a set to specify that each token in the first term will query all the tokens in the second term. We will revise $j'$ to $j$ for clearness.
>
>
> ### **3. Unclear Implementation Details (Respond to R3Q3)**
> LRM's image-to-triplane decoder uses 16 attention heads in all multi-head attention. In the cross-attention layer (details in Section 4.2 and Appendix A.2), only the positional embeddings are applied as Q (queries), and the image features are applied as K and V (keys and values), all being projected to the same dimension $d_{D}=1024$. Then, each of the 32x32x3=3072 positional embedding tokens will query all the 32x32=1024 image feature tokens, resulting in weight shape [32x32x3, 32x32]. Therefore, when multiplied by the Values [32x32, 1024], it produces output tokens of dimension [32x32x3, 1024] that match the input dimension of the next layer.
> We appreciate the reviewer for pointing this out, and we will add the missing details to our paper.
>
>
> ### **4. Volume Rendering (Respond to R3Q4)**
> Our LRM uniformly samples 128 points for each ray in neural rendering. We have also tried uniform sampling with perturbation and two-stage coarse-to-fine sampling but did not see an obvious difference in results. We will leave the deeper investigation of this problem to future work.

---

> > ### Comment · Reviewer_ZEtF · 2023-11-20
> >
> > Thanks for the detailed reply. I have gone through other reviews and all the responses with additional results. I think this work is with high-quality and provides a promising direction. I'll keep my score as strong accept.

---

### Official Review · Reviewer_RnNv · 2023-10-30

**Soundness:** 3 good
**Presentation:** 3 good
**Contribution:** 3 good
**Rating:** 8
**Confidence:** 3

**Summary:**

The paper proposes LRM, a Transformer-based NeRF model that transforms an input image into its corresponding 3D triplane representation.
Compared to recent image-to-3D approaches that rely mainly on post-processing/optimization strategies to extract 3D representations from pre-trained image diffusion models, LRM stands out by directly establishing a 2D-to-3D mapping.

Qualitative results show improved performance in generating novel views and synthesizing unseen objects

----------
Post discussion phase:
I agree with the other reviewers that the paper is a good contribution to the image-to-3D problem, and the authors' responses have addressed all my questions. Therefore, I raised the score from 6 to 8 (good paper, accept).

**Strengths:**

The paper presents an interesting system that hallucinates/synthesizes 3D appearance from DINO features. Compared to the recent methods, LRM excels at
- Directly producing 3D representations in a single forward pass, instead of running optimization to construct a 3D model for each input instance.
- LRM retains details from the input view better, possibly due to the use of image-based features.
- LRM does not require canonicalized training objects, making it easier to apply LRM to other datasets.

Additionally, the paper presents comprehensive ablation studies, showing how each design choice affects the final performance. Overall, the method is presented clearly and easy to follow, the architecture design is sound, and the qualitative results look promising.

**Weaknesses:**

Although the results look promising, the paper has two main weaknesses:
- Insufficient quantitative comparisons: the paper does not conduct any quantitative evaluation against other methods. I believe the novel view synthesis and 3D reconstruction can be evaluated on the held-out sets for those 3D object datasets, and user study should also be possible. Even if the quantitative results may not reflect the generation quality entirely, the paper should include discussions on why these scores are not reliable/not feasible.
- The quality on the occluded side is still limited. For objects that have overall smooth/uniform textures, LRM seems to do a good job filling in the occluded side information. But for more complex, asymmetric patterns (e.g., Figure 2. Giraffe,  supp. website shoe example), LRM struggles to synthesize plausible appearances.

In summary, the absence of quantitative comparisons raises significant concerns.

**Questions:**

My primary concerns stem from the lack of quantitative comparisons. The paper should provide a proper evaluation against other approaches. Even if the standard metrics cannot accurately reflect the qualitative improvement of synthesized/hallucinated objects. The paper should include proper discussions on why the existing metrics are unsatisfactory.

---

> ### Author Response · Authors · 2023-11-16
> **Respond to Reviewer 2 (RnNv)**
>
> We thank Reviewer 2 (RnNv) for acknowledging the contribution of our paper and providing thoughtful comments. Please see our response to the feedback below.
>
> ### **1. Quantitative Comparison (Respond to R2W1 and R2Q1)**
> We thank the reviewer for raising this question, and provide a quantitative comparison to the stat-of-the-art methods Point-E [1], Shap-E [2], and One-2-3-45 [3]. All methods are mentioned in our paper; Point-E trains an image-to-3D point cloud diffusion model, Shap-E encodes point clouds to latent representations and trains a diffusion model on the latents to generate parameters of a 3D implicit function, and One-2-3-45 reconstructs multiview images generated with a 2D diffusion model.
> We randomly selected 100 objects from the Google Scanned Objects (GSO) dataset and measured the novel view synthetic quality of 20 reference views (FID, CLIP-Similarity, PSNR, LPIPS) and the geometric quality (Chamfer Distance), as shown in the Table below.
> We can see that our LRM consistently outperforms previous approaches in all metrics. We will add these results and discussion to our paper.
>
> |    | FID $\downarrow$ | CLIP-Similarity $\uparrow$ | PSNR $\uparrow$ |  LPIPS $\downarrow$ |  Chamfer Distance $\downarrow$ |
> | :---                |   :----:       |    :----:   |      :----:   |     :----:   |    ---: |
> | Point-E        | 123.70  | 0.741  | 15.60  |  0.308  |  0.099  |
> | Shap-E        | 97.05  | 0.805   | 14.36 |  0.289 | 0.085 |
> | One-2-3-45  |  139.24  |  0.713  |  12.42  |  0.448  | 0.123  |
> | LRM (ours)  | **31.44**  | **0.902**  |  **19.60**  |  **0.163**  |  **0.053**  |
>
> [1] Point-E: A System for Generating 3D Point Clouds from Complex Prompts. Nichol et al., 2022.
>
> [2] Shap-E: Generating Conditional 3D Implicit Functions. Jun and Nichol, 2023.
>
> [3] One-2-3-45: Any Single Image to 3D Mesh in 45 Seconds without Per-Shape Optimization. Liu et al., 2023.
>
>
> ### **2. Quality on the Occluded Side (Respond to R2W2)**
>
> We thank the reviewer for pointing out this limitation. As discussed in Section 4.3.2, multiple plausible solutions exist for the occluded side, but our model is deterministic and likely produces averaged modes of the unseens.
> This issue is a joint effect of LRM's deterministic approach and an L2 loss and applying MVImgNet data for training, which mostly only covers 180 degrees of view. A similar issue can be seen in the Masked Autoencoders [1], where the model tends to reconstruct blurry contents when a large block of region is removed from an image. One potential solution is to post-optimize the LRM's output (e.g., using generative prior and SDS loss proposed by DreamFusion [2]) at the cost of extra processing time.
>
> On the other hand, our ongoing research found that LRM can be easily extended to accept sparse multi-view inputs by passing encoded multi-view image features to the image-to-triplane decoder, and making the positional embeddings query from features of all images via cross-attention to construct the triplane. This model can create very high-fidelity shapes where the issue of blurry occluded portions can be largely eliminated due to the higher coverage of the 3D object. We will make the multiview LRM paper public to the research community.
>
> [1] Masked Autoencoders Are Scalable Vision Learners. He et al., 2021.
>
> [2] DreamFusion: Text-to-3D using 2D Diffusion. Poole et al., 2022.

---

> > ### Comment · Reviewer_RnNv · 2023-11-20
> > **Score changed**
> >
> > Thanks for the detailed responses, I have gone through the comments from other fellow reviewers, as well as all the additional results provided. I am quite satisfied, and have changed the scores to reflect this.

---

### Official Review · Reviewer_ZwhW · 2023-10-30

**Soundness:** 4 excellent
**Presentation:** 4 excellent
**Contribution:** 4 excellent
**Rating:** 8
**Confidence:** 5

**Summary:**

The paper presents the first *large-scale* single-view 3D reconstruction method. Given an input image, it uses DINO to extract features from an input image, then uses a *large* transformer with cross-attention and modulation to process these features into triplanes, creating a triplane NeRF, thereby conducting novel view synthesis. The method is trained on large-scale 3D/multiview datasets Objaverse and MvImgNet, with 128 A100 GPUs.

**Strengths:**

* To my knowledge, this paper is the first work showing the scaling ability of transformers on novel view synthesis.
* The task of single-view novel view synthesis is extremely challenging and well-motivated.
* The method is extremely efficient during inference as it only requires a single forward pass, unlike many generative models, e.g. score-based generative models and/or optimization-based methods.
* As a non-generative model, it is astounding for me to see its amazing performances. It seems to be able to handle the ill-posed one-to-many pretty well, despite being a discriminative model.

**Weaknesses:**

* The method requires significant computational resources.
* A minor issue, but the paper shows no quantitative comparison with any prior work.
* Since the method is discriminative, it is not able to sample different realizations of an input. Additionally, the averaging of modes, even though has been weakened a lot compared with prior works such as PixelNeRF, still exists as the author mentioned.
* The task of novel view synthesis is inherently probabilistic, as the author mentioned. Even though the method shows amazing scaling-up ability, it is questionable whether solving a generative task in a discriminative way is reasonable.
* The contribution of this paper comes mainly from its presentation of the ability of large transformers and large-scale 3D data on novel view synthesis. Technically, the efforts mainly come from the engineering efforts combining different techniques and processing with the data.

**Questions:**

* Section 4.3.2 probablistic -> probabilistic.
* Have the authors tried to perform any sparse view reconstruction?

---

> ### Author Response · Authors · 2023-11-16
> **Respond to Reviewer 1 (ZwhW) -- Part 1**
>
> We thank reviewer 1 (ZwhW) for acknowledging the contribution of our paper and providing thoughtful comments. Please see our response to the feedback below.
>
> ### **1. Computational Cost (Respond to R1W1)**
>
> Like many other recent visual/language models (e.g., CLIP [1], GPT-3 [2], and Stable Diffusion [3]), large-scale training is widely applied to stabilize the optimization processes of large models and achieve the best results at the cost of high computational demand.
> Our best-performing LRM was trained on 128 NVIDIA (40G) A100 GPUs for 3 days, but the inference only requires a single 11G GPU.
>
> In addition to the final model that we presented in the paper, we also attempted to train a smaller version of LRM.
> The model uses the same architecture as the baseline model we showed in Appendix C Table 1, and we trained it on 8 NVIDIA (40G) A100 GPUs for 4 days with batch size 64 and gradient accumulation of 4 steps. This model achieves performance between the baseline and the final models shown in Table 1, which is able to produce reasonable results for research purposes.
>
> [1] Learning Transferable Visual Models From Natural Language Supervision. Radford et al., 2021.
>
> [2] Language Models are Few-Shot Learners. Brown et al., 2020.
>
> [3] High-Resolution Image Synthesis with Latent Diffusion Models. Rombach et al., 2021.
>
>
> ### **2. Quantitative Comparison (Respond to R1W2)**
> We thank the reviewer for raising this question, and provide a quantitative comparison to the stat-of-the-art methods Point-E [1], Shap-E [2], and One-2-3-45 [3]. All methods are mentioned in our paper; Point-E trains an image-to-3D point cloud diffusion model, Shap-E encodes point clouds to latent representations and trains a diffusion model on the latents to generate parameters of a 3D implicit function, and One-2-3-45 reconstructs multiview images generated with a 2D diffusion model.
> We randomly selected 100 objects from the Google Scanned Objects (GSO) dataset and measured the novel view synthetic quality of 20 reference views (FID, CLIP-Similarity, PSNR, LPIPS) and the geometric quality (Chamfer Distance), as shown in the Table below.
> We can see that our LRM consistently outperforms previous approaches in all metrics. We will add these results and discussion to our paper.
>
> |    | FID $\downarrow$ | CLIP-Similarity $\uparrow$ | PSNR $\uparrow$ |  LPIPS $\downarrow$ |  Chamfer Distance $\downarrow$ |
> | :---                |   :----:       |    :----:   |      :----:   |     :----:   |    ---: |
> | Point-E        | 123.70  | 0.741  | 15.60  |  0.308  |  0.099  |
> | Shap-E        | 97.05  | 0.805   | 14.36 |  0.289 | 0.085 |
> | One-2-3-45  |  139.24  |  0.713  |  12.42  |  0.448  | 0.123  |
> | LRM (ours)  | **31.44**  | **0.902**  |  **19.60**  |  **0.163**  |  **0.053**  |
>
> [1] Point-E: A System for Generating 3D Point Clouds from Complex Prompts. Nichol et al., 2022.
>
> [2] Shap-E: Generating Conditional 3D Implicit Functions. Jun and Nichol, 2023.
>
> [3] One-2-3-45: Any Single Image to 3D Mesh in 45 Seconds without Per-Shape Optimization. Liu et al., 2023.
>
>
> ### **3. Discriminative vs. Generative Approaches (Respond to R1W3 and R1W4)**
> Similar to the classic PixelNerf [1] and Pixel2Mesh [2] and many other deterministic approaches, LRM considers a simplified assumption of one-to-one mapping between image and 3D. Intuitively, this line of work focuses on reconstructing the average shape given partial 2D information, just as humans can naturally infer a probable 3D model of an object from just a single view. We believe that the definition of single-image-to-3D task is not necessarily generative but depends on the specific application.
>
> One important goal of our paper is to justify the plausibility of large-scale training for 3D, which is very under-explored in previous research.
> Moreover, our LRM learns generic 3D prior, and it bridges 2D and 3D representations that can potentially facilitate 3D generation in the future. For instance, training latent diffusion models on triplanes predicted by LRM to generate new shapes or reconstruct
> 3D models from text-to-multiview generations.
> We would like to refer to the comments from Reviewer 4 (eyyQ), "It fills the gap between 2D and 3D foundation generative model" and "The architecture pave the way for other 3D generative models to scale up".
>
> [1] PixelNeRF: Neural Radiance Fields from One or Few Images. Yu et al., 2021.
>
> [2] Pixel2Mesh: Generating 3D Mesh Models from Single RGB Images. Wang et al., 2018.

---

> ### Author Response · Authors · 2023-11-16
> **Respond to Reviewer 1 (ZwhW) -- Part 2**
>
> ### **4. Averaging Modes and Sparse View Reconstruction (Respond to R1W3 and R1Q2)**
> The issue of averaging modes mainly appears in the texture of the occluded portion of the shapes. It is a joint effect of LRM's deterministic approach and an L2 loss and applying MVImgNet data for training, which mostly only covers 180 degrees of view. A similar issue can be seen in the Masked Autoencoders [1], where the model tends to reconstruct blurry contents when a large block of region is removed from an image. One potential solution is to post-optimize the LRM's output (e.g., using generative prior and SDS loss proposed by DreamFusion [2]) at the cost of extra processing time.
>
> On the other hand, our ongoing research found that LRM can be easily extended to accept sparse multi-view inputs by passing encoded multi-view image features to the image-to-triplane decoder, and making the positional embeddings query from features of all images via cross-attention to construct the triplane. This model can create very high-fidelity shapes where the issue of blurry occluded portions can be largely eliminated due to the higher coverage of the 3D object. We will make the multiview LRM paper public to the research community.
>
> [1] Masked Autoencoders Are Scalable Vision Learners. He et al., 2021.
>
> [2] DreamFusion: Text-to-3D using 2D Diffusion. Poole et al., 2022.
>
>
> ### **5. Paper Contribution (Respond to R1W5)**
> We agree with the reviewer that our proposed LRM is inspired and based on many previous successes in computer vision, such as image representation learning (DINO), inter-modal modeling (transformer-based cross-attention), 3D representation (triplane-NeRF), and the idea of large-scale training. But we do respectfully disagree on "the efforts mainly come from the engineering efforts", as there are enormous alternative design choices, while LRM was built from considerate research insights; our method successfully integrates those components into a scalable and end-to-end trainable system, justifying the plausibility of large-scale 3D training.
> Distinctively, LRM is a feed-forward network that does not rely on any generative prior, and it considers 2D images and 3D shapes as bridgeable modalities and models their correspondence via cross-attention without explicitly defining any spatial alignment.
> LRM is more data-friendly and efficient than recent large-scale 3D models such as Shap-E [1] and Point-E [2], and it is the state-of-the-art single-image-to-3D approach that produces very high-fidelity results.
> We kindly suggest that the methods and ideas presented in this paper carry huge research value that can impact future 3D deep learning studies.
>
> [1] Point-E: A System for Generating 3D Point Clouds from Complex Prompts. Nichol et al., 2022.
>
> [2] Shap-E: Generating Conditional 3D Implicit Functions. Jun and Nichol, 2023.
>
>
> ### **6. Typo (Respond to R1Q1)**
> We thank the reviewer for pointing out this typo; we will fix it in our paper.

---

> > ### Comment · Reviewer_ZwhW · 2023-11-20
> >
> > Thanks to the authors for the rebuttal, I like the paper and will maintain my score.

---

### Meta-Review · Area_Chair_9dv6 · 2023-12-11

**Metareview:**

The submission presents a successful attempt to scale up transformers for novel view synthesis.  The idea is simple, and the results are surprisingly good.  Four expert reviewers appreciated the strong results, including those from the rebuttal, and the clear presentation; they all recommended at least a strong accept.   The AC agrees.

**Justification For Why Not Higher Score:**

N/A

**Justification For Why Not Lower Score:**

Reviewers unanimously recommended strong accepts, acknowledging the (surprisingly) good results and the detailed analyses.

---

### Decision · Program_Chairs · 2024-01-16

Accept (oral)